# Improving Neural Program Synthesis with Inferred Execution Traces

**Richard Shin**[*]
UC Berkeley
ricshin@cs.berkeley.edu

**Illia Polosukhin**
NEAR Protocol
illia@nearprotocol.com

**Dawn Song**
UC Berkeley
dawnsong@cs.berkeley.edu

## Abstract

The task of program synthesis, or automatically generating programs that are consistent with a provided specification, remains a challenging task in artificial intelligence. As in other fields of AI, deep learning-based end-to-end approaches have made great advances in program synthesis. However, compared to other fields such as computer vision, program synthesis provides greater opportunities to explicitly exploit structured information such as execution traces. While execution traces can provide highly detailed guidance for a program synthesis method, they are more difficult to obtain than more basic forms of specification such as input/output pairs. Therefore, we use the insight that we can split the process into two parts: infer traces from input/output examples, then infer programs from traces. Our application of this idea leads to state-of-the-art results in program synthesis in the Karel domain, improving accuracy to 81.3% from the 77.12% of prior work.

## 1 Introduction

The task of *program synthesis* is to automatically generate a computer program that satisfies some specification. It is a problem that has been studied since the earliest days of artificial intelligence [Waldinger and Lee, 1969, Manna and Waldinger, 1975]. With the renewed popularity of neural networks for machine learning in recent years, neural approaches to program synthesis have correspondingly attracted greater attention from the research community. One set of approaches, referred as *neural program induction* by Devlin et al. [2017b], involves learning parameters for a neural network architecture with a design inspired by existing computational structures such as stacks [Grefenstette et al., 2015, Joulin and Mikolov, 2015], random-access and associative memory [Kurach et al., 2016, Graves et al., 2016], and GPUs [Kaiser and Sutskever, 2015].

A different set of approaches, *neural program synthesis*, instead learn to generate explicit discrete programs in a domain-specific language from a specification that consists of as few as 5 input/output example pairs. Several recent papers have proposed neural network-based approaches to *program synthesis* from input/output examples [Parisotto et al., 2017, Devlin et al., 2017b, Bunel et al., 2018]. These methods use an end-to-end encoder-decoder approach, where a neural network learns to generate a program from an encoding of a program specification (a set of input/output examples) from a large synthetic training dataset.

End-to-end approaches like these have been particularly successful in perceptual domains like computer vision where designing intermediate representations is a big challenge. In contrast, within program synthesis, there exists a great deal of structure and auxiliary information the model could learn to exploit in addition to the typically used input/output examples. An example is program execution traces, which have also been used to great effect by prior work [Reed and de Freitas, 2016, Wang et al., 2018].

---

[*]Work partially performed at NEAR.

Given that an execution trace can be a strict superset of an input/output example, intuition suggests that program synthesis from execution traces should be easier than synthesis from I/O examples. Since we can replay an execution trace with the interpreter to obtain detailed information about the program state at each step, a trace-based synthesis model can rely upon the interpreter to handle the semantics of the DSL's operations, and focus more on how to infer control flow constructs by reconciling different paths taken in different inputs. However, execution traces are difficult to obtain as they are much more challenging for the end user to specify, so it is hard to reap their benefits.

In this work, we use the insight that if encoder-decoder neural networks can synthesize programs from input/output examples, they should also be able to infer execution traces. Thus, we can split the problem into two steps: use input/output examples to infer execution traces, and then use execution traces to infer the program. Our empirical results show that this modification leads to state-of-the-art results on the Karel [Pattis, 1981] program synthesis task, improving upon Bunel et al. [2018] from 77.12% to 81.3% accuracy.

Our analysis shows greater accuracy on programs of varying lengths and complexities, demonstrating the general utility of the approach. This is despite the fact that we only use straightforward maximum likelihood training, which is easier to tune than reinforcement learning methods of prior work. Nevertheless, as our method is largely orthogonal to prior techniques like reinforcement learning, our research suggests useful future directions for further improving the accuracy of neural program synthesis.

## 2    Related work

**Program synthesis from examples.**    There have been several practical applications of programming by example based on search techniques and carefully crafted heuristics, such as Gulwani [2011]. More recent work has started to apply deep learning for program synthesis from examples such as RobustFill [Devlin et al., 2017b], DeepCoder [Balog et al., 2016], Neuro-Symbolic Program Synthesis [Parisotto et al., 2017], and Deep API Programmer [Bhupatiraju et al., 2017]. Gaunt et al. [2016] provides a comparison of various program synthesis from examples approaches on different benchmarks, showing limitations of existing gradient descent models. Bunel et al. [2018], which also uses the domain of synthesizing Karel programs from examples, learns to predict the correct program with a deep learning model by leveraging the syntax constraints of the program language and training via reinforcement learning to generate more consistent programs.

**Program induction.**    Another line of recent work aims to teach neural networks the functional behavior of programs, by augmenting the neural architecture with additional computational modules such as Neural Turing Machines [Graves et al., 2014a], Neural GPUs [Kaiser and Sutskever, 2015], and stack-augmented RNNs [Joulin and Mikolov, 2015]. However, these approaches require a large dataset of input-output pairs to learn a single program, and can have trouble generalizing to unseen inputs. An alternative approach involves training neural networks to reproduce execution traces, like in Neural Program-Interpreters [Reed and De Freitas, 2015]. Even though this approach generalizes better, it still requires training a separate model per task. Devlin et al. [2017a] overcomes this weakness by using a meta-learning approach, where the model can induce a program from observing a new task at test time to induce program and produce answer on unseen inputs. Nevertheless, the learned program is induced latently within the weights and activations of a neural network, which limits our ability to interpret which program has been learned and may limit the complexity of programs that the model can represent and execute. Indeed, Bunel et al. [2018], which synthesizes explicit Karel programs, obtained more accurate results compared to Devlin et al. [2017a] which uses an induced latent representation of Karel programs.

**Leveraging interpreters for inverse graphics.**    In recent years, there has been work on learning semantics of interpreters for inverse graphics with neural networks: learning how to control a drawing engine to reproduce a given picture, or in other words, recovering its underlying structure. In Ellis et al. [2017], the authors first infer the execution trace of a drawing program and leverages a generic program search algorithm on the trace. Ganin et al. [2018] instead simultaneously uses techniques from reinforcement learning and adversarial training to teach an agent how to generate a program which renders the desired image. These methods provide evidence that having explicit prediction of

traces or steps aids in learning the semantics of an interpreter, which is an important component of program synthesis.

## 3 Background

$$
\begin{aligned}
\text{Prog } p \quad &:= \quad \texttt{def main():} s \\
\text{Stmt } s \quad &:= \quad \texttt{while}(b):s \mid \texttt{repeat}(r):s \mid s_1;s_2 \\
&\quad \mid \quad a \mid \texttt{if}(b):s \mid \texttt{if}(b):s_1\texttt{else}:s_2 \\
\text{Cond } b \quad &:= \quad \texttt{markersPresent() } \mid \texttt{leftIsClear()} \\
&\quad \mid \quad \texttt{rightIsClear() } \mid \texttt{frontIsClear() } \mid \texttt{not}(b) \\
\text{Action } a \quad &:= \quad \texttt{move() } \mid \texttt{turnLeft() } \mid \texttt{turnRight()} \\
&\quad \mid \quad \texttt{pickMarker() } \mid \texttt{putMarker()} \\
\text{Cste } r \quad &:= \quad 0 \mid 1 \mid \cdots \mid 19
\end{aligned}
$$

Figure 1: The syntax of the Karel DSL as used in this paper. Figure from Devlin et al. [2017a].

**Problem domain.** Karel is an educational programming language (Pattis [1981]), used for example in Stanford CS introductory classes and the Hour of Code initiative. It features an agent inside a grid world, where certain cells can contain *markers* or *walls* (but not both); the agent cannot enter cells where there is a wall. The agent can take the following actions: moving forward (`move`), turning left or right (`turnLeft, turnRight`), and modifying the world state by removing or adding *markers* to the current location (`pickMarker, putMarker`).

Karel programs, which are imperative, can contain branching statements (`if, ifElse`), `while` loops which execute as long as a condition is true, and `repeat` loops which execute for a fixed number of repetitions. The following conditions are available: whether the cell at the agent's location contains markers (`markersPresent`), and whether there are any walls nearby (`frontIsClear, leftIsClear, rightIsClear`).

Devlin et al. [2017a] introduced the use of the Karel domain for program *induction*, where a neural network learns to represent a program; in this work, we tackle Karel program *synthesis*. The goal in this domain is to learn how to generate a program in the Karel DSL given a small set of input and output grids. Formally, we are given a set of $n$ input-output world pairs $\{(I_1, O_1), \cdots, (I_n, O_n)\}$, with some hidden program $\pi$ which satisfies the property that executing $\pi$ in $I_1$ results in $O_1$, $I_2$ results in $O_2$, and so on. Our task is to recover $\hat{\pi}$ by observing the $n$ input/output pairs, such that $\hat{\pi}$ is semantically equivalent to $\pi$; in other words, $\hat{\pi}$ should have the same effect as $\pi$ on any input world, but they do not need to be textually equivalent. However, note that the problem is under-specified: with $n$ input-output pairs, it is not possible to disambiguate among all possible $\pi$, so the model needs to pick the most promising among the possibilities.

Our work is based on Bunel et al. [2018], which applied a neural encoder-decoder approach to Karel program synthesis, similar to the work of Devlin et al. [2017b] and Parisotto et al. [2017] which was for a string-editing domain. Bunel et al. [2018] used both supervised learning with a randomly-generated synthetic dataset to train their model, as well as a reinforcement learning-based approach to further improve the model's program synthesis accuracy. As part of their work, they have developed a deep learning architecture for Karel program synthesis which we use as the basis for our approach.

## 4 Approach

### 4.1 Motivation

Past work in program synthesis, program induction, program repair, and other areas of machine learning have explored the benefits of learning using *execution traces*.

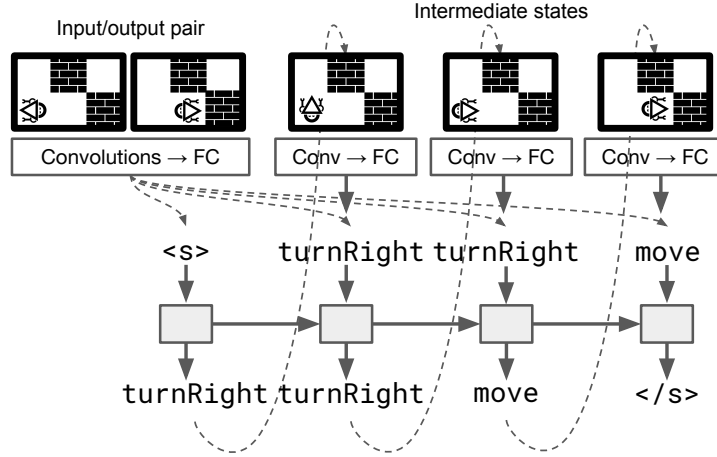

Figure 2: Architecture of I/O → TRACE model.

For example, in program induction, the Neural Programmer-Interpreter [Reed and de Freitas, 2016] receives execution traces as supervision, and can learn complex tasks more quickly and accurately; given recursive traces, it can exhibit perfect generalization [Cai et al., 2017]. Other program induction models such as the Neural Turing Machine [Graves et al., 2014b] and Neural GPU [Kaiser and Sutskever, 2016] have the harder task of learning directly from input/output examples, and thus need a very large amount of training data and careful hyper-parameter tuning. In program repair, Wang et al. [2018] improve upon baseline methods by learning models that use execution traces. Ellis et al. [2017] and Ganin et al. [2018] generate execution traces for the purposes of inverse graphics. Even for generative modeling of images, learning to generate the picture incrementally and additively (similar to execution traces in our setting) has been able to improve performance [Gregor et al., 2015].

We hypothesize that any program synthesis model must be able to internally reason about the semantics of the DSL and in particular the atomic operations. For example, in the Flash Fill system [Gulwani, 2011], this knowledge is explicitly specified by the system's creators. In contrast, neural program synthesis methods need to learn both the semantics of the DSL and how to synthesize programs in the DSL entirely from the training data.

Even beyond the past work using traces and our hypothesis above, our intuition as programmers suggests that it should be easier to synthesize a Karel program given not only the input/output examples, but also the list of steps taken by the correct program to transform the input into the output. However, it is impractical to expect an end user specifying the desired program to also provide the correct execution trace for the program, since devising the trace is almost as much work as writing the actual program itself.

However, if neural networks can learn to synthesize programs from input/output examples as shown by Bunel et al. [2018] and others, it follows that they should also be able to synthesize the execution trace from input/output examples. Indeed, we can expect this to be an easier problem given that the execution trace does not contain any control flow constructs. Furthermore, as the model iteratively generates the execution trace, we can evaluate the partial trace with the Karel interpreter and provide its internal state to the model to help guide the model's next output.

Once we have a model that can recover the correct execution trace from the input/output examples for a desired program, it becomes possible to train and use a program synthesis model that takes both input/output examples and corresponding execution traces as the program specification. By including information extracted from the Karel interpreter as we run the execution trace about the current state of the world at each point in the trace, the program synthesis model has less need to internally reason about the program's semantics as some of that work is effectively offloaded to the Karel interpreter.

In the subsequent sections, we detail how we built two models to split the Karel program synthesis problem ("I/O → CODE") into two parts: I/O → TRACE, then TRACE → CODE.

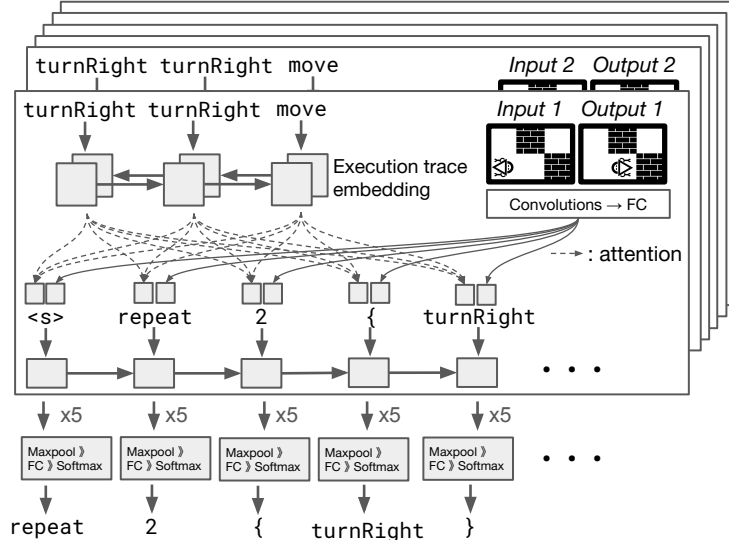

Figure 3: Architecture of TRACE → CODE model. We follow the architecture in Bunel et al. [2018], but add an execution trace input.

## 4.2 Predicting execution traces from input/output pairs

In this work, an execution trace refers to an ordered set of actions: $(action_1, ..., action_T)$. In the case of Karel, actions are $move$, $turn\{Right, Left\}$, $\{put, pick\}Marker$. For a given original training example $(\pi, \{(I_1, O_1), ..., (I_N, O_N)\})$, we can generate $N$ training examples $(I_1, O_1, (action_1, \cdots, action_T)_1), ..., (I_N, O_N, (action_1, \cdots, action_T)_N)$ for trace prediction by running $\pi$ on these $I/O$ pairs and recording the actions taken by the program. Thus, if the original training data contained $K$ examples where each example contained $N$ I/O pairs, then we obtain a I/O pair to trace dataset of $K \cdot N$ examples suitable for supervised learning.

Figure 2 shows the deep learning model architecture we used for this task. To encode the input/output examples, we use a convolutional neural network with a final fully-connected layer, taken from Bunel et al. [2018]. To generate the sequence of actions, we use a two-layer LSTM decoder. At each step of the decoder, it receives as input the concatenation of the following: 1) an embedding of the action taken in the previous step, 2) the input/output pair embedding, and 3) an embedding of the current state of the grid after executing all of the past actions. In theory, the LSTM can learn to keep track of the current state of the grid internally, rendering the third input unnecessary; however, as we will see in Section 5.4, explicitly using the Karel interpreter to track the current grid state helps the model better understand how the grid is changing after each action and maintain more context.

## 4.3 Synthesizing programs from input/output examples and execution traces

To create a model which uses both a set of input/output examples and execution traces for generating the desired program, we started with the architecture from Bunel et al. [2018] and extend it to also take the execution trace as an input. Specifically, we add a bidirectional LSTM to the architecture which is responsible for producing an embedding of the execution trace for each step in the trace.

Given an execution trace $(action_{1,n}, ..., action_{T,n})$ and an initial state $state_{1,n} := I_n$ for the $n$th input/output example, we can use the Karel interpreter to replay the actions to obtain $state_{2,n}, \cdots, state_{T+1,n}$. For Karel, each state contains the full grid world and the objects within it: the location of the agent, its orientation, the current number of markers in each cell, and the locations of walls (which cannot be manipulated by the agent and therefore do not change through the course of execution). Given the mismatch in lengths, and also given that the actions semantically occur in between the states, we interleave the two to create an input of length $T + (T + 1) = 2T + 1$: $(state_{1,n}, action_{1,n}, state_{2,n}, \cdots, action_{T,n}, state_{T+1,n})$.

To provide this input to the bidirectional LSTM, we embed $state_{t,n}$ and $action_{t,n}$ in the following way. For each state, we evaluate the four conditionals (`markersPresent`, `frontIsClear`, `leftIsClear`, `rightIsClear`) that can influence the program's flow of execution, embed each boolean value and concatenate them. For each action, we look up the corresponding embedding from a table. Both sets of embeddings are randomly initialized and learned. We will denote the $2T + 1$ outputs for the $n$th trace from the bidirectional trace LSTM as $\mathcal{T}_{1,n}, \cdots, \mathcal{T}_{2T+1,n}$.

We also tried a variant where the input consists of $T + 1$ elements. First, we append appending a final `</s>` to the list of actions: $action_{T+1,n} = $ `</s>`, to make the lengths match. We then embed each $state_{t,n}$ and $action_{t,n}$, and then concatenate the embeddings together. In addition to the conditional values, we also tried providing an embedding of the grid itself to the LSTM, using a similar convolutional neural net as used to encode each grid in the I/O → TRACE model. However, we found that both variants were inferior compared to the method described in the previous paragraph.

Following Bunel et al. [2018], we also encode each input/output example using a convolutional neural network with a final fully-connected layer. We generate the program one token at a time with a decoder LSTM. As in Bunel et al. [2018] and Devlin et al. [2017b], we run a separate LSTM for each input/output pair; the LSTMs have separate states but shared weights. At decoding step $i$, the LSTM for the $n$th input/output example receives the concatenation of following:

- an embedding of the token generated in step $i - 1$, or of `<s>` at step 0 (the start of decoding)
- the I/O pair embedding for $(I_k, O_k)$,
- the *context vector* $c_{i-1,n}$ for step $i - 1$: a weighted sum of $\mathcal{T}_{1,n}, \cdots, \mathcal{T}_{2T+1,n}$, computed using a multiplicative attention mechanism based on $o_{i-1,n}$, the LSTM's output at step $i - 1$. Note that $c_{0,n} = \mathbf{0}$.

We obtain the decoder LSTM output $o_{i,n}$ for each of the $N$ input/output pairs, compute the context vector $c_{i,n}$ as described above, and concatenate them to obtain $\tilde{o}_{i,n} = \text{Concat}(o_{i,n}, c_{i,n})$. To compute the logits over the $i$th program token, we compute $W \cdot \text{MaxPool}(\tilde{o}_{i,1}, \cdots, \tilde{o}_{i,N})$, where $W \in \mathbb{R}^{v \times d}$, and $v$ is the size of the output vocabulary. See Figure 3 for a visual depiction of the overall architecture.

To train this model, we tried two different sources of supervision. Recall that each entry in the provided training data consists of a program $\pi$ and 5 input/output pairs $\{(I_1, O_1), ..., (I_5, O_5)\}$. First, we can execute $\pi$ on $I_1, \ldots, I_5$ to obtain the execution trace on each input; we refer to this trace as the *gold* trace. However, we will not have access to the gold trace when we wish to actually use this model for program synthesis from input/output examples.

Second, we can use the I/O → TRACE model from Section 4.2 to infer a valid trace for the given I/O pair; we refer to this trace as the *inferred* trace. Unfortunately, this model does not always succeed at recovering a correct trace for the I/O pair, in which case we substitute a trace containing a single `UNK` action and two grid states, the input grid and the output grid. Furthermore, the trace may deviate from the actions taken by the true program even if the final state is identical, as certain actions can be permuted without any effect on the output (such as `turnLeft` and `pickMarker`), and certain sequences of actions (such as `turnLeft` then `turnRight`) are no-ops. Nevertheless, we will only have access to the inferred trace at inference time, so it is useful to match the training and test distributions more closely.

## 5 Experiments

### 5.1 Training dataset and procedure

To train and test our models, we used the same dataset as Bunel et al. [2018], from `https://bit.ly/karel-dataset`. The training dataset consists of 1,116,854 entries, and the test dataset contains 2,500 entries. Each entry in the dataset contains a Karel program and 6 input/output pairs which satisfy that program. For training the I/O → TRACE model, we used all 6 input/output pairs within each entry for a total of 6,701,124 training traces. For training the TRACE → CODE model (and our reimplementation of the I/O → CODE model from Bunel et al. [2018]), we randomly sample 5 out of the 6 input/output examples (and corresponding traces) each time we sample an entry from the training data. In general, we endeavored to follow the training regime from the prior work as closely as possible, although we discovered that SGD with gradient clipping worked better for training the models than Adam. For all of the evaluations of TRACE → CODE we used beam search with size 50.

Table 1: Comparison of our best model with previous work from Bunel et al. [2018]. "Gen." stands for generalization accuracy.

| | Top-1 | | Top-50 | |
| | Exact Match | Gen. | Guided Search | Gen. |
|---|---|---|---|---|
| MLE [Bunel et al., 2018] | 39.94% | 71.91% | – | 86.37% |
| RL_beam_div_opt [Bunel et al., 2018] | 32.17% | 77.12% | – | 85.38% |
| I/O → CODE, MLE | 40.1% | 73.5% | 84.6% | 85.8% |
| I/O → TRACE → CODE, MLE | **42.8%** | **81.3%** | **88.8%** | **90.8%** |

## 5.2 Performance metrics

When evaluating the model, we use beam search both to get outputs that have higher log likelihood than what can be obtained with greedy decoding, and also to obtain multiple candidate sequences. As such, we use multiple criteria to report the performance of the models.

First, for purposes of comparison, we have the same metrics as used by the prior work: **Top-K Exact Match**, which measures how often one of the top K output programs of the model textually matches the original program exactly; and **Top-K Generalization**, which denotes the fraction of test instances for which one of the top K output programs will have the correct behavior across the 5 input/output examples used to specify the program to the model, as well as the held-out 6th input/output example.

As an alternative to these metrics, we suggest to use what we call **Top-K Model-Guided Search Accuracy**. In this metric, we consider the top K program outputs in order, from most likely to least likely. We test each candidate program on the 5 input/output examples that specify the program, and see if it works correctly on those 5. We return the first such program (the top-ranked one) as the solution, and then test it on the held-out 6th program to report the accuracy. The motivation for this approach is two-fold. First, we already have the 5 I/O examples to specify the program for the model to produce, and so we might as well use them to filter any unsatisfactory outputs of the model, to reap the benefits of having a precisely checkable specification for the correct answer. Second, this metric is more comparable to other methods in the literature that use a search-based method for program synthesis, either with handwritten heuristics or with machine learning models (such as [Balog et al., 2016]); indeed, such methods will often try thousands or millions of candidate programs, rather than the comparatively small $K = 50$ which we used for our experiments.

## 5.3 Evaluation of I/O → TRACE → CODE

In Table 1, we compare our best I/O → TRACE → CODE model (created by gluing together I/O → TRACE and TRACE → CODE) against the previous work of Bunel et al. [2018]. We reimplemented their MLE model (labeled as I/O → CODE), obtaining slightly better results compared to theirs.

We note that we did not implement the RL_beam_div_opt training method of Bunel et al. [2018], and so our results are all based on MLE training. Nevertheless, our I/O → TRACE → CODE method outperforms all others on all metrics, including the best result in Bunel et al. [2018]. We anticipate that using reinforcement learning methods (such as RL_beam_div_opt) can improve our method's accuracy even further.

We also analyzed how models performed on various slices of the test data in Table 2: programs with no control flow (only actions); programs with conditionals (`if` or `ifElse`) but not loops (`repeat` or `while`); programs with loops but no conditionals; and programs containing at least one control flow element. We also partitioned the data depending on the length of the gold program into three buckets.

We can observe that I/O → TRACE → CODE improves upon I/O → CODE within every slice of the data. The magnitude of the improvement is most significant on long programs, which provides supporting evidence for our hypothesis in that the I/O → CODE model would need to internally keep track of the Karel state but have trouble doing so.

Table 2: Comparing performance on different slices of data

| Slice | % of dataset | I/O → CODE | I/O → TRACE → CODE | Δ% |
|---|---|---|---|---|
| No control flow | 26.4% | 100.0% | 100.0% | +0.0% |
| Only Conditions | 15.6% | 87.4% | 91.0% | +3.6% |
| Only Loops | 29.9% | 91.3% | 94.3% | +3.0% |
| With all control flow | 73.6% | 79.0% | 84.8% | +5.8% |
| Program length 0-15 | 44.8% | 99.5% | 99.5% | +0.0% |
| Program length 15-30 | 40.7% | 80.8% | 86.9% | +6.1% |
| Program length 30+ | 14.5% | 48.6% | 61.0% | +12.4% |

Table 3: Evaluation of I/O → TRACE models.

| | Top-1 | | Top-10 | |
|---|---|---|---|---|
| | **Exact Match** | **Correct** | **Exact Match** | **Correct** |
| No grids | **58.7**% | 92.5% | **59.3**% | 95.9% |
| LGRL | 57.6% | 94.8% | 58.0% | 97.4% |
| **PRESNET** | 57.8% | **95.2%** | 58.2% | **98.0%** |

## 5.4 Evaluating I/O → TRACE and TRACE → CODE separately

For the first part of our approach (I/O → TRACE), in Table 3 we show results of predicting the 5 execution traces from the 5 input/output examples used to specify a Karel program synthesis task. In this table, we consider a result to be correct if all 5 predicted traces transform the corresponding input states to the output states when executed in the Karel interpreter. As discussed in Section 4.2, there exists many possible execution traces which transform a given input state to the output state; therefore, the exact match accuracy is much lower than the correctness metric.

The LGRL model uses the same architecture for convolutional encoder as in Bunel et al. [2018]. We also augmented it with residual connections between layers, results for which are reported as PRESNET model. Furthermore, to confirm that the interpreter's current state helps the I/O → TRACE model produce correct traces, we have trained a variant ("No grids") which omits the inputs of the grid state.

For the second part (TRACE → CODE), Table 4 compares a model trained on *gold* traces against one trained on *inferred* traces from the best I/O → TRACE model. Due to the distributional differences between the gold and inferred traces, the model trained on gold traces does poorly on inferred traces.

We also tried an evaluation using the gold execution traces from the test set. As discussed earlier in Section 4.3, the gold execution traces would not normally be available at test time, so this evaluation serves as a hypothetical comparison against our main result.

## 6 Discussion and Future Work

From our results, we consider confirmed our hypothesis that it is beneficial to use traces for explicitly training a model that learns the semantics of interpreter, separately from the task of synthesizing the code for the correct program.

Interestingly, the predicted trace and gold trace fail to match exactly in half of the cases even though the predicted trace is correct. Indeed, the TRACE → CODE model trained on gold traces doesn't perform as well on inferred traces. That said, when we evaluated TRACE → CODE on gold traces at validation time, it outperformed the model trained on inferred traces. Since I/O → TRACE independently predicts traces for each I/O pair, we hypothesize that they lack consistency with each other compared to the gold traces. Thus, for future work we suggest to investigate inferring the execution trace for a given I/O pair, conditioned on already generated execution traces for the same underlying program but on different I/O pairs. We also noticed that the TRACE → CODE

Table 4: Evaluation of TRACE → CODE models.

| Train traces | Test traces | **Exact Match** | **Correct** | **Guided Search** |
|---|---|---|---|---|
| Gold | Inferred | 39.2% | 76.5% | 81.8% |
| Inferred | Inferred | **42.8%** | **81.3%** | **88.8%** |
| Gold | Gold | 54.0% | 86.4% | 92.4% |

model trained on predicted traces performed much worse when evaluated on gold traces compared to predicted traces. This phenomenon suggests that training a TRACE → CODE model with multiple options of traces sampled from I/O → TRACE and from gold traces may improve the model's resilience and further improve the accuracy.

We also leave for future work exploring usage of reinforcement learning objectives (similar to Bunel et al. [2018]) for the training of these models. We see two possible ways of applying these objectives: training I/O → TRACE and TRACE → CODE models separately with the reward of passing unseen tests; and training I/O → TRACE and TRACE → CODE models jointly end-to-end, where traces are decoded into symbolic form and reinforcement learning allows propagation of learning signals between the two parts.

## Acknowledgements

This material is in part based upon work supported by Berkeley Deep Drive, the National Science Foundation under Grant No. TWC-1409915, and the Defence Advanced Research Projects Agency under Grant No. FA8750-17-2-0091. Any opinions, findings, and conclusions or recommendations expressed in this material are those of the authors and do not necessarily reflect the views of the above organizations.

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
