[Reviews · NeurIPS 2018]

Reviewer 1



Summary: This paper provides a novel program synthesis approach by inferring the intermediate states: execution trace. Unlike conventional methods that treat program synthesis as an end-to-end task based on input/output (i/o) examples, in this manuscript, the authors attempt to infer the execution trace from i/o examples, and further synthesize the program given the i/o and execution trace. The motivation is clear and straightforward since generating programs from execution traces should be easier. Experiments on Karel program synthesis task show the effectiveness of the proposed approach. The research question is significant and very hard. It is good to see more program characteristics are included in the design of the neural network. Strengths: 1. Though straightforward, the proposed framework for program synthesis is novel. It splits the program synthesis into two steps by introducing the execution traces. 2. The experiments are well-designed and the results are superior to the competing methods greatly. 3. This paper is well-organized and the writing is good. Weaknesses: 1. A few parts of this paper are not very clear or not sufficiently provided, such as the model details. Section 4.3 should be addressed more to make it clearer since some concepts/statements are misleading or confusing. 2. The trace to code model is complex, which requires as many LSTMs as the number of input/output pairs, and it may be hard to be applied to other program synthesis scenarios. 3. Apart from the DSL, more experiments on dominant PL (e.g., Python) would be appreciated by people in this research field. Details are described as below: 1. For a given program, how to generate diverse execution traces that can be captured by the I/O -> Trace model? Since execution traces are generated by running the program on N I/O pairs, it is possible that some execution traces have a large overlap. For example, in the extreme case, two execution traces may be the same (or very similar) given different I/O pairs. 2. The authors involve the program interpreter in their approach, which is a good trial and it should help enhance the performance. However, I am curious about is it easy to be integrated with the neural network during training and testing? 3. The concept of state is not very clear, from my understanding, it represents the grid status (e.g., agent position) and it is obtained after applying an action of the trace. Line 186-line 187, is the “elements” equivalent to “states”? or “actions”? More should be elaborated. 4. Line 183-line 184, is it necessary to use embedding for only four conditionals (of Boolean type)? only 16 possible combinations. 5. As depicted in Figure 3, if more I/O pairs are provided, the Trace->Code should be very complex since each i/o example requires such an LSTM model. How to solve this issue? 6. In essence, the Trace->Code structure is a Sequence to Sequence model with attention, the only differences are the employment of I/O pair embedding and the max pooling on multiple LSTM. How are the I/O pair embeddings integrated into the computation? Some supplementary information should be provided. 7. It is interesting to find that model trained on gold traces perform poorly on inferred traces, the authors do not give a convincing explanation. More exploration should be conducted for this part. 8. It would be better if some synthesized program samples are introduced in an appendix or other supplementary documents.

Reviewer 2



This paper proposes a practical extension to neural program synthesis by leveraging program execution traces in addition to the I/O pairs associated with example Karel program. The authors reason that by providing explicit execution traces from Karel interpreter, learning algorithm has less to internally reason about specific mapping from input to output. Furthermore it simplifies the learning problem given that the execution trace does not contain any control flow constructs. The proposed deep learning model has a two-stage approach (I/O -> Trace and Trace-> Code). This approach is shown to perform better than a recent one-stage I/O-> Code baseline (Bunel 2018) on different types of programs (specifically longer and complex programs). The paper will be of practical interest to programming community. It is well-written, and experiment analysis is adequate. The novelty is limited in my opinion as execution trace is a standard programming construct and it is not surprising that addition of such feature improves overall performance.

Reviewer 3



The authors propose to break down the neural program synthesis pipeline into two steps: a model that maps IO examples to corresponding execution traces, and a model that given a set of traces generates a corresponding program. The model is tested on the dataset of [Bunel 2018] and slices thereof and compares favorably to the baseline. Pros: - The paper is well written (except for the model description, which is not really formal enough. thankfully code will supply what the paper is lacking.) - The related work is satisfactory - The idea is catchy and charmingly simple; the fact that it frees the model from having to keep track of the internal state of the interpreter is appealing - The comparison to the baseline is favorable I think that this is a fairly good empirical paper, worthy of acceptance. Minor issues: - It is always unfortunate when there is only one baseline, but I do agree that comparing to methods developed for different settings or languages (e.g. the end-to-end differentiable prover of Riedel et al.) may be difficult in neural program synthesis - In the related work section most articles are missing